# Back to Reality for Imitation Learning

**Edward Johns**
The Robot Learning Lab at Imperial College London

**Abstract:** Imitation learning, and robot learning in general, emerged due to breakthroughs in machine learning, rather than breakthroughs in robotics. As such, evaluation metrics for robot learning are deeply rooted in those for machine learning, and focus primarily on data efficiency. We believe that a better metric for real-world robot learning is time efficiency, which better models the true cost to humans. This is a call to arms to the robot learning community to develop our own evaluation metrics, tailored towards the long-term goals of real-world robotics.

**Keywords:** Imitation learning, Reinforcement learning, Evaluation, Benchmarks

## 1 Introduction

**How do we define the term *robot learning*?** Let us consider two options. Firstly, *the study of machine learning methods for applications to robotics*. And secondly, *the study of methods to enable robots to learn*. The choice between these two can be distilled down to whether robot learning is considered fundamentally to be a field within machine learning, or a field within robotics, respectively. But whilst this distinction may appear to be arbitrary, the implication is in fact profound, for the following reason. It determines the evaluation metrics used by the community to judge the performance of new methods, and therefore, it determines the long-term impact that the field is shaping itself around. In this paper, we argue that metrics which guide development of a technology should be driven by the target application of that technology, irrespective of the methods used by that technology. And therefore, we argue that robot learning metrics should be driven by real-world robotics applications, rather than the underlying machine learning algorithms. However, this is not what we observe in the robot learning community today: evaluation is dominated by traditional machine learning metrics. In this paper, we expose some problems with this, which deserve foresight, attention, and debate.

**Robot learning emerged from machine learning, not robotics.** Whilst the fields of both machine learning and robotics have existed for decades, the field of robot learning is a more recent development, such as with the arrival of our first CoRL in 2017. But its growth in popularity can largely be attributed to breakthroughs in machine learning, and in particular, breakthroughs in deep learning and reinforcement learning, rather than breakthroughs in classical robotics. And during this emergence, the robot learning community has brought with it evaluation metrics that were designed by the machine learning community, for machine learning problems. For example, a typical reinforcement learning paper published at a machine learning conference today, will present graphs of *Success Rate vs Number of Environment Interactions*. And we see these same graphs dominating papers published at CoRL. Now, for the machine learning community, this metric is sensible: it encourages data efficiency, which is arguably the most fundamental aim in machine learning research. But is data efficiency the most fundamental aim in robotics research?

## 2 Back to Reality

**The curse of simulators.** To answer the above question, we now examine the differences between the simulation benchmarks typically evaluated on in machine learning papers, and the real-world environments of robotics applications. Our first observation is that, whilst simulation benchmarks often assume prior existence of a task definition, such as a reward function, in reality each new task must be manually defined by the human teacher. And if we are aiming for robots to learn from

Blue Sky Papers, 5th Conference on Robot Learning (CoRL 2021), London, UK.

everyday humans in everyday environments, rather than from engineers coding hand-crafted reward functions, then this should be an intuitive and natural process, such as a human demonstration of the task. As such, when we move away from simulators and into the real world, a large number of robot learning problems can be considered as a form of imitation learning. This is particularly the case for robot manipulation, where the huge diversity of tasks makes it impractical to manually define each one in code. Our second observation is that, whilst simulators facilitate rapid prototyping and large-scale quantitative benchmarking, this comes at the cost of making two problematic assumptions: (1) environment resetting comes for free, and (2) low-dimensional ground-truth states (e.g. object poses) are readily available. In real-world robotics, however, (1) automatic environment resetting requires significant human supervision or construction of task-specific apparatus, and (2) obtaining low-dimensional states requires additional development of a state estimator for each new task.

**From data efficiency to time efficiency.**   The above observations lead to our main argument: data efficiency should not be the most fundamental aim in robot learning research, and instead, we should be optimising our methods for *time efficiency*. We define time efficiency as how quickly a robot can learn a new task in a period of time, regardless of the amount of data collected during that time, and we argue that time is a much better approximation of the true cost to human teachers. Time has historically been itself approximated by the amount of data required, but as we have observed, this approximation is limited. If a method requires regular environment resetting, then the additional time to reset the environment should be a penalty, when that method is evaluated relative to other methods that require less environment resetting. Similarly, if a method requires object poses, then the additional time taken to train that pose estimator should be accounted for. But this is not to say that data efficiency should not be encouraged; data efficiency is always beneficial as long as it does not come at the cost of time efficiency. Instead, we argue for a more comprehensive metric than only data efficiency: the best imitation learning method is one which optimally trades off all aspects that contribute to time efficiency, and data efficiency is just one of those aspects. We believe that this new metric will encourage researchers to develop methods which are more useful to humans in practice, when moving away from the simulators and the lab, and back into the real world.

**Two flavours of time efficiency.**   Imitation learning involves humans teaching robots. This takes time, but it also takes effort, and an ideal imitation learning method would be one which minimises both time and effort. However, these factors are not independent. For example, a reinforcement learning method seeded with a larger number of demonstrations may learn more quickly, but at the cost of more human effort. As such, we now propose two flavours of the time efficiency metric.

*Clock time efficiency.* We define clock time efficiency as the speed at which a robot learns a new task with respect to the amount of wall clock time. This could be the metric of choice in a factory setting, for example, where an engineer would like a robot to learn a new task as quickly as possible to then begin a manufacturing job, even if the engineer needs to supervise the learning throughout.

*Human time efficiency.* We define human time efficiency as the speed at which a robot learns a new task with respect to the amount of time a human spends supervising the learning. This could be the metric of choice in a domestic setting, for example, where a consumer would like the robot to learn a new task as autonomously as possible and with minimal human effort, even if the robot takes a long time to learn.

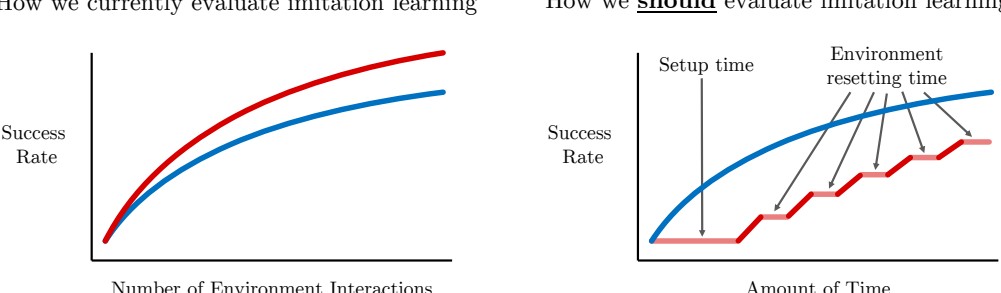

Figure 1: A comparison of two hypothetical methods using data efficiency (left) and time efficiency (right) metrics.

**A choice of two futures.** Figure 1 left shows how robot learning methods are typically evaluated today. According to the data efficiency metric, the red method is superior. But hypothetically, let us now consider that the blue method requires no environment resetting and learns end-to-end from images, whilst the red method requires regular environment resetting and ground-truth object poses. Figure 1 right then shows how these two methods would be evaluated with a time efficiency metric. Here, whilst the data efficiency of the blue method is proportional to its time efficiency, the red method is significantly less time efficient than it is data efficient. Today, we are being encouraged to develop the red method. Through this paper, we aim to motivate the robot learning community towards a future where we are instead developing the blue method.

## 3    Meta-analysis of CoRL 2020

**Methodology.** Having established our criteria for good imitation learning metrics, we can now take a look at evaluation methods used by the robot learning community today. To do this, we performed a meta-analysis of imitation learning papers published at CoRL 2020. To select these papers, we first searched for all papers which contained either "imitation" or "demonstration" in the paper's abstract. From these, we inspected each paper and retained those which actually involved providing a demonstration to learn a new task, which resulted in 19 papers. We then analysed each paper to determine whether it evaluated success rate as a function of either clock time or human time. We also recorded whether a paper assumed ground-truth object poses, and whether it required environment resetting. In some cases, the experiments themselves did not require environment resetting but only because the tasks were simple target reaching tasks, and in these cases we made a judgement as to whether environment resetting would be required with more typical, everyday tasks. For each paper, we also recorded whether any real-world experiments were done.

**Results.** Table 1 shows the results of this meta-analysis. Papers are grouped into 8 groups, where all the papers in one group contained the same yes/no answers across all 5 columns. The first observation we make is that not a single paper evaluated the clock time efficiency or human time efficiency, with most methods instead evaluating data efficiency. Some papers did involve studying success rate as a function of the number of demonstrations, but in those cases, the overall clock time or human time was not evaluated. The second observation we make is that 10 out of the 19 papers assumed access to ground-truth object poses, and 13 papers required regular environment resetting. Neither of these properties make for practical, real-world imitation learning. To obtain these poses, some papers required manually pre-training an object pose estimator for each new task [1, 2], but timing information for this was not provided nor included when comparing the method to alternatives. A final observation we make is that 8 out of the 19 papers only include simulation experiments, an unfortunate trend at all CoRL conferences which results in authors focussing mainly on machine learning metrics, rather than those more appropriate to real-world robotics.

| # Papers | Clock Time Efficiency? | Human Time Efficiency? | Assumes Object Poses? | Environment Resetting? | Real-World? |
|---|---|---|---|---|---|
| 4 [3, 4, 5, 6] | No | No | Yes | Yes | No |
| 3 [7, 8, 2] | No | No | Yes | Yes | Yes |
| 3 [9, 10, 11] | No | No | No | Yes | No |
| 3 [12, 1, 13] | No | No | No | Yes | Yes |
| 3 [14, 15, 16] | No | No | Yes | No | Yes |
| 1 [17] | No | No | No | No | Yes |
| 1 [18] | No | No | No | No | No |
| 1 [19] | No | No | No | No | Yes |

Table 1: Meta-analysis of 19 imitation learning papers at CoRL 2020.

## 4    Moving Forwards

**Simulators are still important.** We have established our argument for the use of time efficiency instead of data efficiency as an evaluation metric. But in practice, both clock time efficiency and human time efficiency are difficult to objectively evaluate with real-world experiments. For example, if the authors themselves are the robot teachers, then there are likely to be implicit or explicit biases in the time taken, depending on whether the authors' own method, or a baseline, are being

evaluated. As such, we still recommend the use of simulation benchmarks for large-scale objective evaluation, but with some significant changes that will mitigate the aforementioned problems with these benchmarks. Specifically, these changes should model any aspects of task learning which require time to perform. For example, each time an environment would be manually reset, a fixed amount of time should be added to both the method's clock time and it's human time counters. As another example, each task can optionally be provided with ground-truth object poses, but a fixed amount of time must then be added to the method's clock time counter, to model the user or robot collecting data for an object pose estimator. Given benchmarks such as these, we believe that the community will begin to develop methods which more optimally trade off all the different aspects of real-world robot learning.

**Visualising time efficiency.**    In Figure 2, we present various options for visualising these metrics, on hypothetical data. Figure 2a compares three different methods by plotting success rates as a function of human time. Whilst Method D has the highest human time efficiency, it is not necessarily the most data efficient. Therefore, it would not necessarily be considered a desirable method if traditional machine learning metrics were used. Figure 2b then presents a more complete evaluation of one individual method. Here, each line represents a particular success rate and trades off the amount of human time required, and the overall clock time required. This is a useful visualisation for choosing an operating point during deployment, where a user may have a particular preference for minimising overall training time, compared to minimising overall human effort or expertise. Figure 2c then shows the trade-off between pre-training time and fine-tuning time across three different methods. For example, meta-learning methods typically require a large amount of pre-training, whereas behavioural cloning methods typically learn from scratch. Here, each line represents one success rate (e.g. the amount of time needed for 90 % success rate). This graph provides an intuitive way to compare methods across entirely different families of imitation learning approaches, which is typically not possible when considering only one of these dimensions. Here, Method I outperforms both Methods G and H, and we can see that Method H is faster than Method G at learning new tasks when there is an abundance of pre-training time, whereas Method G is better when learning from scratch.

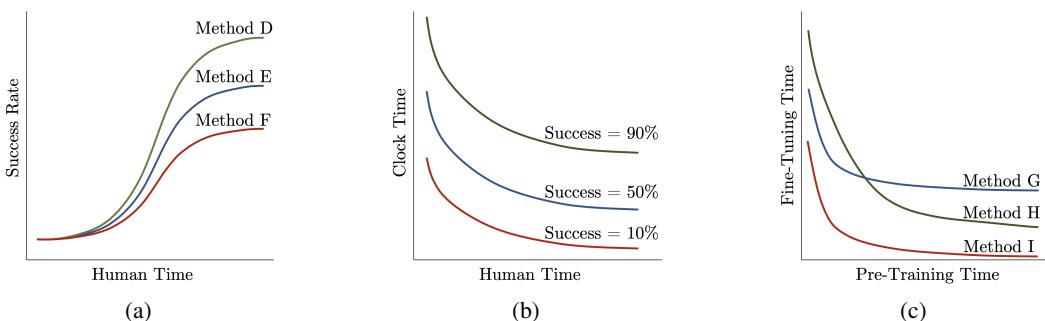

Figure 2: Examples of how evaluations could be visualised using our proposed clock time efficiency and human time efficiency metrics.

## 5    Conclusions

In this paper, we have argued that data efficiency is not the optimal evaluation metric for robot learning, when the long-term goal is real-world robotics. Instead, two alternative aspects should be considered: the overall time taken for learning a new task, and the amount of time spent by the human supervising the learning. Methods which are the most data efficient may not necessarily be the most time efficient, and may sacrifice consideration of real-world practicalities in an attempt to beat the state-of-the-art on simulation benchmarks. Our intention through this paper, is to stimulate the community to develop benchmarks which more accurately reflect the costs to humans in real-world imitation learning. Re-thinking evaluation in the real-world, outside of simulation environments, has inspired our own recent work on imitation learning: from just a single human demonstration, with just a single environment reset, and without any prior object knowledge [20, 21].

**Acknowledgments**

This work was supported by the Royal Academy of Engineering under the Research Fellowship scheme.

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
