# OpenReview forum: "Back to Reality for Imitation Learning"
_robot-learning.org/CoRL/2021/Conference/Blue_Sky — CoRL 2021, Blue Sky_

### Official Review · Reviewer_qtVE · 2021-08-13

**Novelty:** Good
**Impact:** 3
**Clarity Of Presentation:** Very Good

**Recommendation:**

Weak Accept: I recommend accepting the paper, but will not argue for my recommendation if the majority of other reviewers have a different opinion.

**Summary:**

Summary:

The paper argues (with evidence based on past CoRL publications) that the field of imitation learning focuses on metrics (i.e., data efficiency) which are not aligned well with bottlenecks in practice. The paper argues that the field should focus more on metrics which more closely correlate with the actual effort it would take to perform an imitation learning algorithm in the real-world -- e.g., how many human demonstrations are needed, how hard is it to collect those demonstrations, how many manual resets need to be performed during training, etc. The paper proposes to account for all these challenges in new evaluation metrics: "clock time efficiency" and "human time efficiency". The paper provides some conceptual examples of how this metrics can be reported (Fig 1), allowing for more actionable conclusions from future publications.



**Summary Of Recommendation:**

Reactions:

I generally agree with the paper's overall theme. Certainly many of CoRL's papers rely too much on simulation and have limited connection to real-world bottlenecks. I think proposing metrics more aligned with these real-world bottlenecks is a step in the right direction. Although, I wonder if another potential direction -- proposing new tasks -- could be a more effective way to do this. The paper does not discuss this approach.

Another concern I have with the proposal in the paper is the lack of concrete definitions for human/clock-time efficiency. These are mostly conceptually introduced with a few examples. Even in the Fig 1 plots, we see these conceptual metrics realized as several different concrete metrics: "human time", "pretraining time", "fine tuning time". I understand that different settings may require different definitions, but at the same time I fear that if we don't define a concrete and specific set of metrics, we will end up with each paper evaluating on their own variant of the metric, and thus limit reproducibility and comparison across papers. Perhaps a better way to implement the paper's proposal is via a "checklist" or set of questions, similar to the "reproducibility checklist" which has appeared as part of other conferences' reviewing process (I recall NeurIPS 2020 had such a checklist).

---

### Official Review · Reviewer_dK76 · 2021-08-25

**Novelty:** Good
**Impact:** 3
**Clarity Of Presentation:** Very Good

**Recommendation:**

Weak Reject: I recommend rejecting the paper, but will not argue for my recommendation if the majority of other reviewers have a different opinion.

**Summary:**

This paper conducted a meta-analysis of imitation learning papers published at CoRL 2020 and examined the evaluation metrics to measure the effectiveness of their algorithms. The authors argued that "time efficiency" instead of "data efficiency" would be a better alternative to quantify the true cost of imitation learning algorithms for real-world robotic problems. To support the proposed time efficiency metric, the authors examined 19 publications from CoRL 2020 on imitation learning and investigated the metrics and experimental setups of these papers. In the end, the authors advocated two aspects of time efficiency to be considered for real-world robotics, including the overall time taken for learning a new task and the amount of time spent by the human supervising the learning.

**Summary Of Recommendation:**

The paper is written nicely. The authors started with a thought-provoking discussion on the definition of "robot learning," the recent boom of robot learning rooted in machine learning, and finally, the metrics used by robot learning algorithms. The focus on designing the appropriate evaluation metrics is crucial to ensure that our community is making consistent and sustainable progress. The discussion on imitation learning is particularly meaningful among the robot learning algorithms, as they have played an increasingly important role in designing autonomous robots' behaviors.

This paper argued that data efficiency is not the best metric reflecting the cost of robot learning algorithms and proposed using clock time and human time instead. While the reviewer agreed with the overall sentiment that more realistic metrics should be designed to evaluate learning algorithms for robotics, the use of time might have the following foreseeable issues:

1) Clock time is contingent on the current hardware and simulation technologies. It is possible that the physical simulation utilized by an algorithm could be orders of magnitude faster by new simulation machinery and new computing hardware.
2) Human time efficiency can vary from person to person, depending on their skills, and certain human skill levels might not be easily quantifiable. In addition, human time does not capture the cognitive loads it costs humans in the learning process.
3) It is unclear what should be included (or excluded) when reporting the time efficiency. Does setup time or infrastructure building count? Shall we only count recurring time costs as one-time costs that could be amortized over the robot's lifetime?

The reviewer believed that these issues are important to discuss when designing the time efficiency metrics. However, most of them have not been touched upon in this paper. While the reviewer agreed with the authors that better metrics should be added alongside the commonly used data efficiency metric, it was unclear how this newly proposed time efficiency could be rigorously evaluated and compared across different problem settings and different algorithms. For this reason, the reviewer is leaning towards rejecting the current version of this manuscript.

---

### Decision · Program_Chairs · 2021-10-01

**Decision:**

Accept

**Comment:**

This short paper examines the evaluation metrics for robot learning and argues that the usual notion of sample complexity in ML does not sufficiently capture the effort of gathering data and performance for robot learning. Both reviewers acknowledge the importance of this question. The proposed alternatives (wall clock time, human time, ...) may have limitation, but they present interesting opportunities for discussion and debate. The authors are encouraged to address some of the concerns raised by the reviewers and acknowledge the limitations of the proposed alternatives.